# Experimental Research into the Ultrasonic P-Wave Velocity of Coal Slime Based Backfill Material

Baifu An [1,2,*], Chenghao Cui [1], Jinfang Ren [3], Dongda Wang [4], Jiale Wang [1] and Qiaomei Yi [1]

[1] Work Safety Key Lab on Prevention and Control of Gas and Roof Disasters for Southern Coal Mines, Hunan University of Science and Technology, Xiangtan 411201, China; 21020101080@mail.hnust.edu.cn (C.C.); 22020101013@mail.hnust.edu.cn (J.W.); 20010101004@mail.hnust.edu.cn (Q.Y.)

[2] School of Resource & Environment and Safety Engineering, Hunan University of Science and Technology, Xiangtan 411201, China

[3] Shandong Energy Group Northwest Mining Co., Ltd., Xi'an 710018, China; wangchao@shandong-energy.com

[4] Intelligent Equipment Research Institute of Limited Liability Companies of Changsha Mining Research Institute, Ltd., Changsha 410012, China; wangdongda@zgwk1.wecom.work

\* Correspondence: 1010097@hnust.edu.cn

**Abstract:** To study the influences of different mix proportions of materials on the ultrasonic p-wave velocity of coal slime based backfill materials, influences of four factors on the p-wave velocity of specimens were investigated by designing orthogonal tests and taking coal slime as the principal raw material. The four factors included the following: (A) the mass concentration of coal slime, (B) the content of high-water-content material, (C) cement content, (D) and content of fly ash. Test results revealed the following: (1) the ultrasonic p-wave velocity is in the range of 3.916 to 8.319 km/s and various factors are listed in a descending order as A, D, B, and C according to their influences on the ultrasonic p-wave velocity; (2) the ultrasonic p-wave velocity is positively correlated with the compressive strength and shear strength, with correlation coefficients separately of 0.87 and 0.65; (3) the equations for variations in the ultrasonic p-wave velocity under influences of different factors are fitted. The ultrasonic p-wave velocity has a quadratic polynomial relationship with factor A, while following exponential relationships with factors B, C, and D. A predictive model for characteristic parameters of the ultrasonic p-wave velocity of coal slime based backfill materials jointly influenced by the four factors was established based on the fitting equation for variations of single factors and ultrasonic p-wave velocity. The predictive model was then verified.

**Keywords:** coal slime; backfill material; correlation coefficient; ultrasonic wave speed; formula fitting

## 1. Introduction

At present, the finite nature of coal resources has become an issue of note and the demand for mining engineering to use eco-friendly mining technologies has increased. In such a context, the application of coal slime based backfill materials, as environmental-friendly cemented backfill materials, into coal mines has received wide attention [1]. Coal slime based cemented backfill materials are formed using cementing materials to solidify coal slime, which can be used as the main raw material. Coal slime based cemented backfill materials are generally applied to mine filling engineering during coal mining, to fill mined-out areas and abandoned roadways, stabilize the surrounding rock, and improve the safety factor of mining and the rate of resource utilization. They lay a solid foundation for safe, efficient coal mining [2–6].

It is always a difficulty in project implementation to monitor and evaluate the strength characteristics of coal-tailings based backfill materials in real-time, practical, engineering applications. Conventional mechanical test methods commonly take a long time and call for many resources, and it is challenging to achieve real-time monitoring in the mining environment. Therefore, there is an urgent need to develop a novel method that can rapidly

and accurately assess the strength characteristics of coal-tailings based backfill materials [7–9]. Ultrasonic detection has been widely applied to the evaluation and monitoring of the mechanical properties of materials because it is non-intrusive, highly accurate, and works in real-time. By measuring the propagation velocity of ultrasonic waves in materials, ultrasonic detection can determine mechanical parameters including the elastic modulus and Poisson's ratio of materials, thus providing an important reference for the strength characteristics of materials. However, for materials with complex porous structures, such as coal-tailings based backfill materials, the relationships of the propagation characteristics of ultrasonic waves with the internal structures and mechanical properties remain to be investigated [10–13].

The proportions and properties of cementing materials may affect the density and strength of backfill materials, thus affecting the measurement of wave velocity. Generally, a high proportion of cementing materials will induce a high wave velocity, because a large amount of cementing materials can improve the strength and density of the backfill materials. Different proportions of cementing materials cause the ultrasonic wave velocity to change in different manners. Moreover, the influence of the mix proportion of cementing materials is associated with those of other factors on the measurement of the ultrasonic wave velocity [14–18]. Variation of one factor may change other factors, thus influencing measurement results of the ultrasonic wave velocity. Through experiments and theoretical analysis, this research aimed to expound the influencing mechanism of mix proportions of a coal slime based backfill material on the ultrasonic wave velocity [19–21]. In addition, the association of mix proportions with the material strength was revealed and the corresponding prediction method was proposed. This research provides a novel and efficient strength assessment method for coal mines and solid technological support for the safe and stable operation of coal mines [22,23].

## 2. Experimental Procedures

The main material used in the experiments was coal slime discharged in the coal washing process from Shuiliandong coal mine, Binxian County, Shaanxi Province, China. Three materials, namely, the high-water-content materials, cement, and fly ash were also selected as additives to prepare the cemented backfill material together with coal slime. Therein, the cement was a 42.5# ordinary Portland cement; the fly ash was the secondary fly ash discharged from a power plant. Preliminary tests showed that specimens showed favorable strength characteristics and economy when they were prepared following these mix proportions: mass concentration of coal slime was 28.75% to 40.00%, and the proportions of the high-water-content materials, cement, and fly ash remained at 2.4% to 3.9%, 3.3% to 10%, and 3.3% to 10% of the total mass.

### 2.1. Characteristics of Experimental Raw Materials

2.1.1. Coal Slime

Coal slime [24] is a by-product in the coal washing process, and its main constituents are coal and chemical materials retained in the coal-purification process, including silicon oxide ($SiO_2$), aluminum oxide ($Al_2O_3$), sulphates, ammonium, and heavy metals. Coal slime, which generally contains fine particles and has a large specific surface area, is easily oxidized in air. With strong water retention capability and a high water-content, coal slime generally contains more than 20% water after dehydration. The ash content is commonly high, so coal slime is likely to be clustered and has a low calorific value; because coal slime generally contains many clay minerals, along with its high water content and fine particle size, most coal slime is highly viscous and has certain flowability. When stacking coal slime, the shape of coal slime is extremely unstable and the coal slime is washed away when it comes into contact with water while it flies upwards when it is dried, so coal slime is apt to pollute the environment. Considering this, the stacking, storage, and transportation of coal slime face some difficulties, and the direct use of coal slime is of low economic benefit. The main chemical constituents of coal slime are listed in Tables 1 and 2.

**Table 1.** Elemental analysis of coal (%).

| $C_{ad}$ | $H_{ad}$ | $N_{ad}$ | $S_{t, ad}$ | $O_{ad}$ |
|---|---|---|---|---|
| 38.29 | 2.88 | 0.25 | 0.20 | 8.03 |

**Table 2.** Composition of coal slime ash (%).

| $SiO_2$ | $Al_2O_3$ | $CaO$ | $Fe_2O_3$ | $K_2O$ | $MgO$ | $Na_2O$ | $TiO_2$ |
|---|---|---|---|---|---|---|---|
| 59.68 | 17.58 | 2.37 | 4.10 | 1.96 | 1.18 | 1.22 | 1.51 |

Data in Table 2 are the relative contents of various main minerals in coal slime. Analysis of these constituents is of importance for understanding the material properties of coal slime and its applications in industrial production.

$SiO_2$, as the primary component of coal slime, accounts for almost 60% of the total mass. $SiO_2$ is generally present in many minerals and rocks and exerts significant influences on the viscosity and structural stability of coal slime. The high-content $SiO_2$ imparts high hardness and compressive strength to coal slime.

The $Al_2O_3$ content in coal slime is 17.58%, so $Al_2O_3$ is also an important component. $Al_2O_3$ is commonly seen in many rocks and minerals and may affect the structural strength and fire-resistance of coal slime.

Figure 1 indicates that the particle size of coal slime is in the range of 100 to 500 μm. It is inferred that coal slime shows the following physical properties:

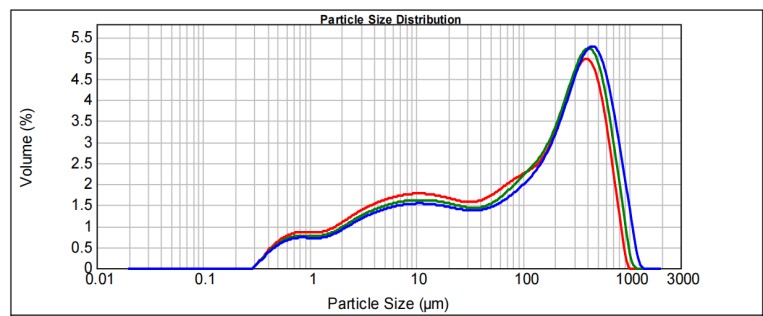

**Figure 1.** The particle size distribution of coal slime. The different colored lines represent measurements of the particle size of the material at different times.

① Flowability and plasticity: coal slime with particle size in this range generally retains a certain flowability, so that it is easy to flow in the filling process and therefore fill voids in mines. Meanwhile, coal slime is also plastic and can be formed into the requisite shape by applying a certain pressure;

② Deformability: coal slime may be deformed to a certain extent after filling (and in particular under pressure);

③ Density: coal slime particle assemblies have small interstitial gaps, so they can form relatively compact filling bodies and improve the stability and bearing capacity of filling bodies.

2.1.2. Characteristics of High-Water-Content Materials

The high-water-content materials [25–27] included two parts, namely, materials A and B. Therein, material A was the mixture of sulfoaluminate cement clinker, suspending agent, and super-retarder, while material B was the mixture of plaster, lime, suspending agent, and fast-setting hardening accelerator. The primary and basic composition of material A was sulfoaluminate cement clinker, while that of material B was plaster. The high-water-content materials showed the characteristic whereby the slurry prepared with each of the materials alone was non-solidified while the slurry prepared with the mixture of the two materials could be solidified rapidly. Based on these characteristics, adding

sulfoaluminate in the coal slime based cemented backfill materials can accelerate the rate of solidification of materials and also provide a certain strength to the cementing materials. The main chemical constituents of the high-water-content materials [27–29] A and B are listed in Tables 3 and 4.

**Table 3.** Chemical composition of high-water-content material A (mass fraction) %.

| $SiO_2$ | $Al_2O_3$ | $Fe_2O_3$ | $SO_3$ | $TiO_2$ | $K_2O$ | $Na_2O$ | CaO | MgO |
|---|---|---|---|---|---|---|---|---|
| 11.85 | 29.73 | 42.52 | 2.55 | 2.93 | 6.92 | 1.00 | 0.34 | 0.05 |

**Table 4.** Chemical composition of high-water-content material B (plaster) (mass fraction) %.

| CaO | $Al_2O_3$ | $Fe_2O_3$ | $SO_3$ | $SiO_2$ | MgO | $K_2O$ | $Na_2O$ |
|---|---|---|---|---|---|---|---|
| 31.32 | 1.34 | 0.50 | 37.05 | 4.42 | 2.97 | 0.26 | 0.08 |

As shown in Figures 2 and 3, the particle size of the high-water-content materials is mainly in the range of 1 to 100 μm. Within that range, the materials are likely to have the following physical properties:

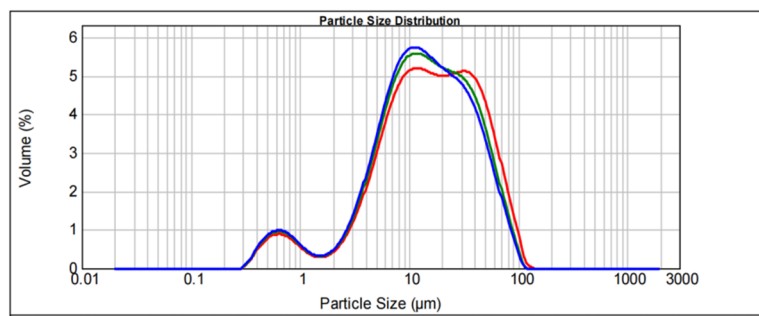

**Figure 2.** The particle size distribution of material A (sulfoaluminate).

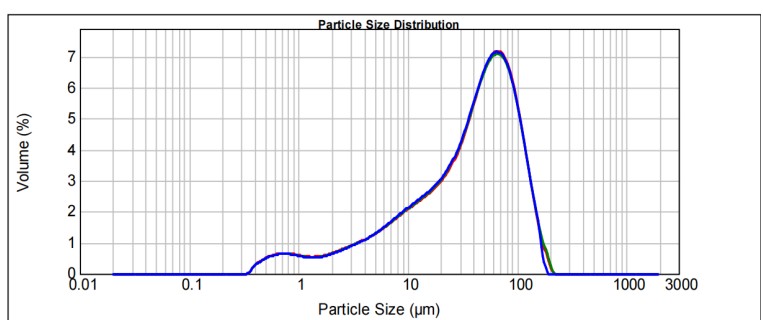

**Figure 3.** The particle size distribution of material B (plaster).

① Ultrafine particles: the particles of high-water-content materials are generally at the micron scale. Therefore, the interaction between particles is significantly affected by surface effects;

② Large surface area: due to the fine particles, the high-water-content materials have a very large specific surface area, facilitating their participation in chemical reactions and improving the activity of materials;

③ High hygroscopicity: due to the fine particles and large specific surface area, the high-water-content materials show favorable hygroscopicity and can adsorb and retain a relatively large content of water, thus reducing the rate of bleeding from filling bodies.

### 2.1.3. Characteristics of Cement

The 42.5# ordinary Portland cement was selected as the additive of the coal slime based backfill material. The cement is able to improve the strength, stability, and durability of filling bodies, thereby guaranteeing the safety and efficiency of coal mining. The chemical composition of the cement is summarized in Table 5.

**Table 5.** Chemical composition of cement (mass fraction) %.

| $SiO_2$ | $Al_2O_3$ | $Fe_2O_3$ | CaO | MgO | $SO_3$ | $K_2O$ | $Na_2O$ | $TiO_2$ |
|---|---|---|---|---|---|---|---|---|
| 22.5 | 5.3 | 4.7 | 61.7 | 2.1 | 2.2 | 0.8 | 0.4 | 0.3 |

These chemical compositions react with each other in the specimen-preparation process, participate in the hydration reaction of hardened cement, and finally form solid filling structures. In the meantime, the contents and proportions of these chemical compositions also affect the physical properties, chemical properties, and engineering applications of cement. Hence, controlling and understanding the chemical composition of the cement is key to the optimal selection and use of cement.

As shown in Figure 4, the particle size of cement mainly ranges from 10 to 50 μm, in which the material may show the following physical properties:

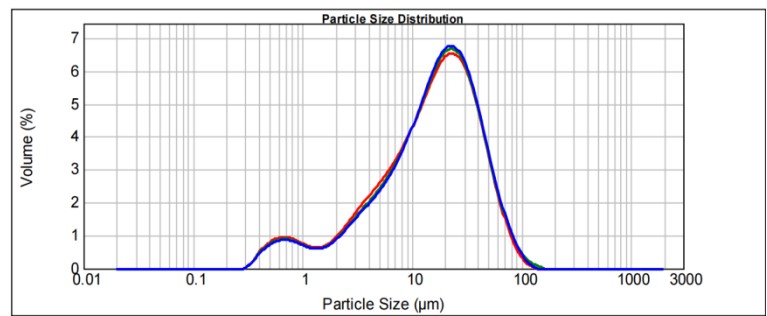

**Figure 4.** The particle size distribution of cement.

① Favorable dispersity: the micro-particles in cement are of high dispersivity when preparing the filling slurry and are not likely to be clustered;

② Adsorbability: due to the large specific surface area, cement particles can adsorb some materials, such as water and chemical compounds, which exerts certain influences on the solidification of the coal slime based backfill material;

③ Active reaction: the micro-particles in cement are of high reactivity, participate in the hydration reaction, and form hardened cement of high strength.

### 2.1.4. Characteristics of Fly Ash

Fly ash is generated in the combustion of coal and is an important material used to fill voids left after mining. Fly ash is mainly sourced from solid particles in the emissions associated with the combustion of coal in coal-fired power plants. To avoid the environmental pollution of discharged fly ash, fly ash was collected as an additive of the coal slime based backfill material to enhance the strength and stability of the filling bodies, so that they can bear geological stress. The chemical composition of fly ash is displayed in Table 6.

**Table 6.** Chemical composition of fly ash (mass fraction) %.

| $SiO_2$ | $Al_2O_3$ | $Fe_2O_3$ | CaO | MgO | $SO_3$ | $K_2O$ | $Na_2O$ | $TiO_2$ |
|---|---|---|---|---|---|---|---|---|
| 55.8 | 22.1 | 11.9 | 3.4 | 2.3 | 3.2 | 0.6 | 0.3 | 0.4 |

$SiO_2$, $Al_2O_3$, and $Fe_2O_3$ generally account for large proportions by mass of fly ash. This exerts important influences on the strength and stability of the specimens of backfill

materials prepared with fly ash and helps to improve the early strength and fire resistance thereof.

It can be seen from Figure 5 that the particle size of fly ash is mainly between 10 and 80 μm, in which the material shows the following physical properties:

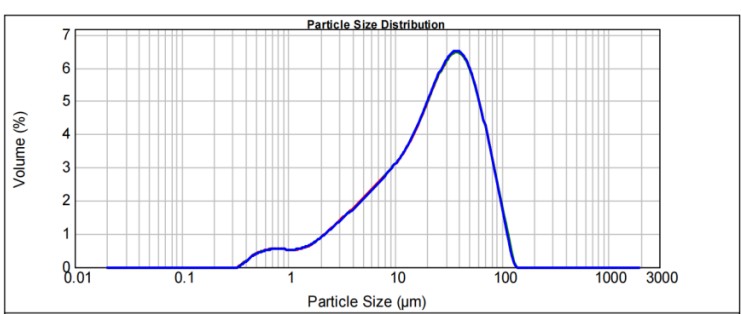

**Figure 5.** The particle size distribution of fly ash.

① An ability to fill voids and stability: the fly ash with an appropriate particle size range can fill in voids in the coal slime based filling bodies, enhance the stability, and avoid the collapse of mined-out areas;

② Inhibiting the dispersivity: some active constituents in fly ash can suppress the dispersion of harmful materials and play a role in protecting the environment around mines;

③ Lowering the cost: use of fly ash, as a by-product, can reduce the cost of backfill materials and improve the economics of engineering/mining operations.

### 2.2. Specimen Preparation Procedures

When weighing materials, it was necessary to guarantee that the container was dry and that it was reset before each weighing. The material A, material B, water, cement, fly ash, and coal slime powder were separately weighed according to the mix proportions in the experiments. The weighed experimental materials were mixed. The operator ensured uniform stirring and avoided loss of water in the stirring process.

① Weighing, stirring, and loading in the mold: the uniformly stirred coal slime based cemented backfill materials were poured into molds, during which the molds were gently shaken. A layer of oil was uniformly smeared on the inner surface of the molds before loading the materials, so as to ensure smooth separation of specimens from molds during demolding;

② Vibration and demolding: to ensure compactness of specimens, the prepared specimens should contain as few bubbles as possible. The loaded molds were gently placed on a flat vibrator to be vibrated for 30 s. After flatting the surfaces of the molds after vibration, the molds were put into a curing chamber to be cured for 7 d and then demolded. The specimens in the early curing period have little strength, so the specimens should be handled gently in the demolding process to avoid damage;

③ Curing: the demolded specimens were placed in a YH-60B curing box to be cured under standard conditions at a temperature of 25 °C and relative humidity exceeding 98%. The entire specimen preparation process in the experiments lasted for 7 d. Standard specimens were prepared according to the nine mix proportions of the materials, and five specimens were prepared under each mix proportion, so a total of 45 standard specimens were prepared. All specimens were cured in a standard curing chamber before experiments.

The specimen preparation process is shown in Figure 6.

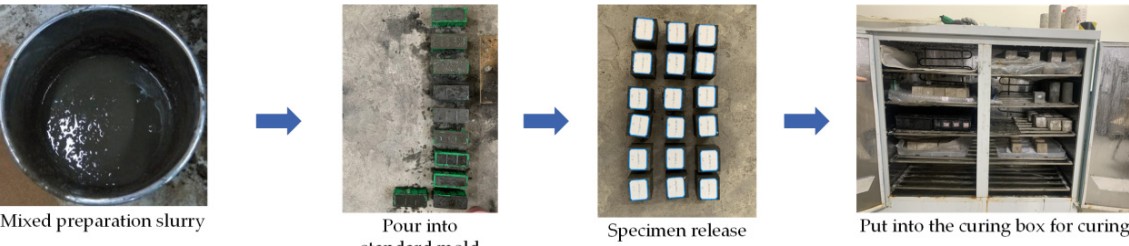

**Figure 6.** Specimen preparation procedures.

## 3. Orthogonal Test Schemes and Test Results

### 3.1. Orthogonal Design of Experiment

The main material used in the tests was coal slime discharged during coal washing in Shuiliandong coal mine. In addition, high-water-content material, cement, and fly ash were selected as additives, which were used together with coal slime to prepare the cemented filling materials. In addition, cement was a 42.5# ordinary Portland cement, and fly ash was secondary fly ash discharged from a power plant. The tested raw materials and scanning electron microscope (SEM) images are shown in Figures 7–11. Previous research has shown that specimens have favorable strength and cost when they are prepared with coal slime at a mass concentration of 28.75~40.00%, with the high-water-content material, cement, and fly ash being 2.4~3.9%, 3.3~10%, and 3.3~10% of the total mass.

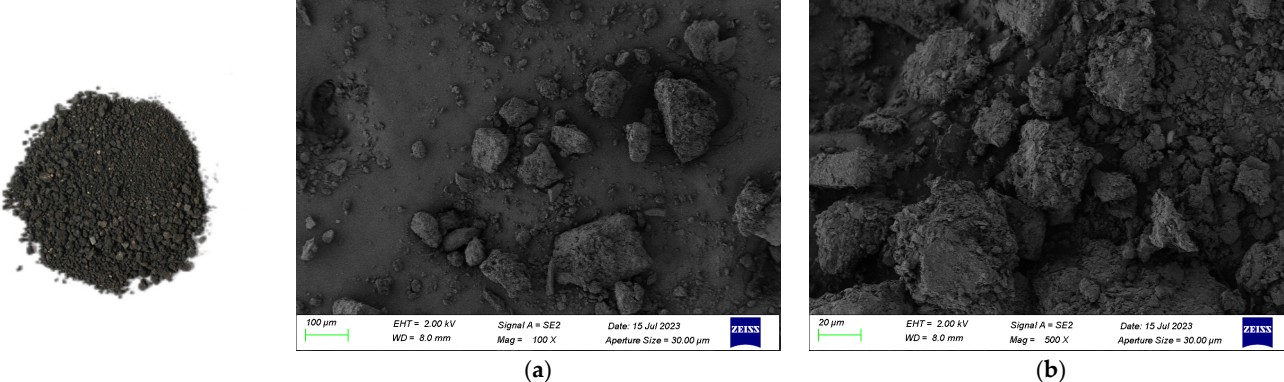

(**a**)          (**b**)

**Figure 7.** Coal slime and SEM images. (**a**) 100×; (**b**) 500×.

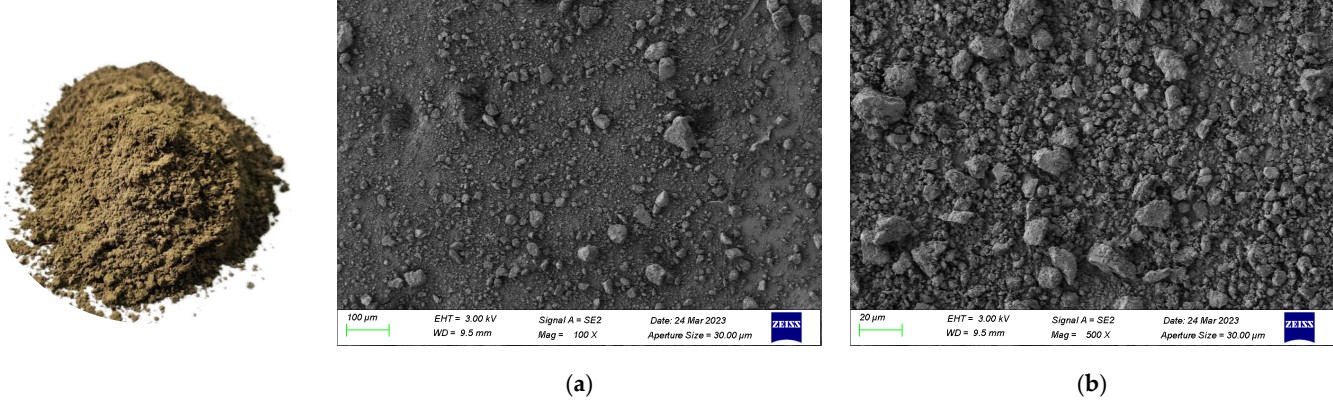

(**a**)          (**b**)

**Figure 8.** High-water-content material A and SEM images. (**a**) 100×; (**b**) 500×.

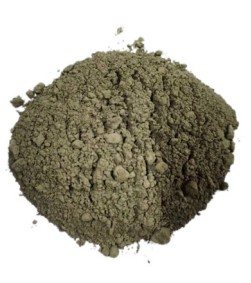

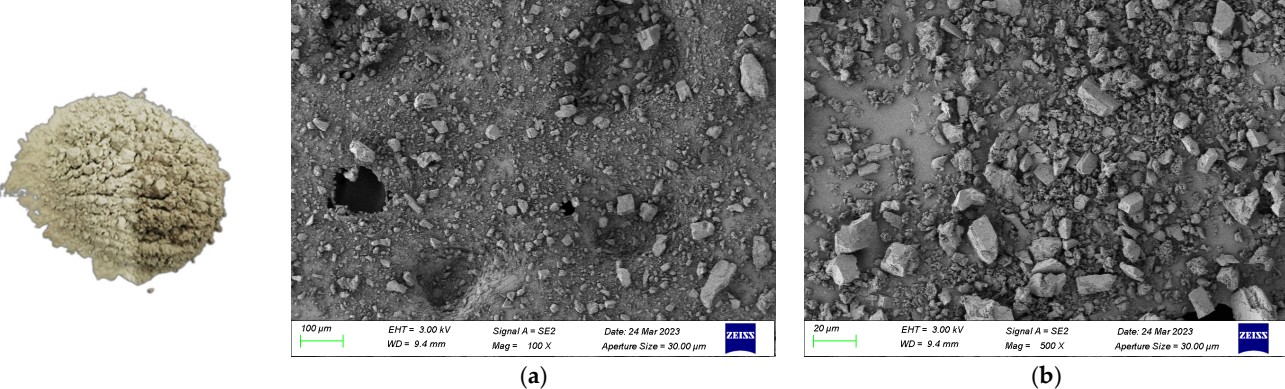

**Figure 9.** High-water-content material B and SEM images. (**a**) 100×; (**b**) 500×.

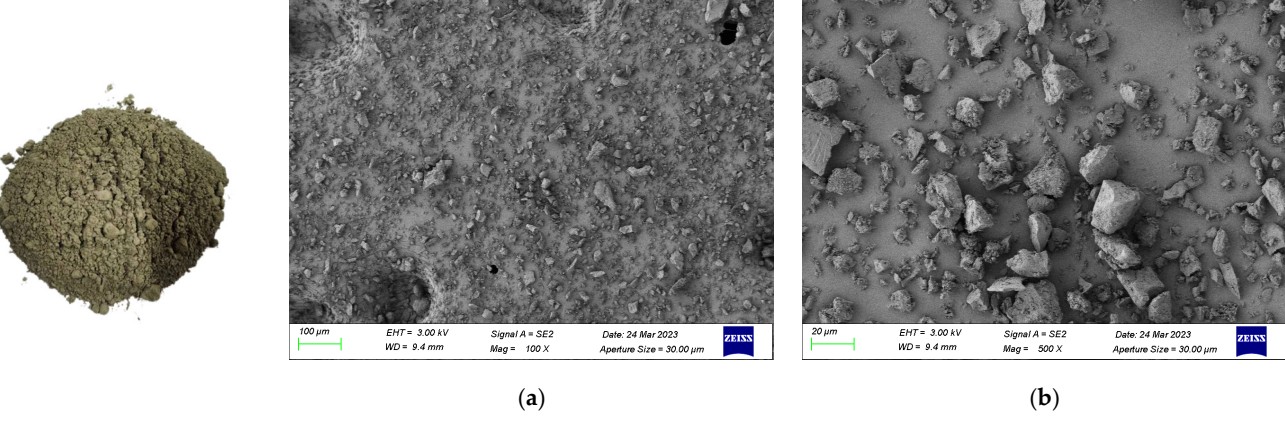

**Figure 10.** Cement and SEM images. (**a**) 100×; (**b**) 500×.

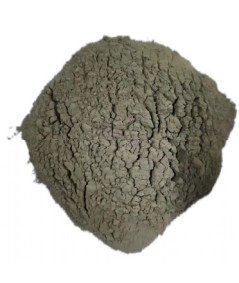

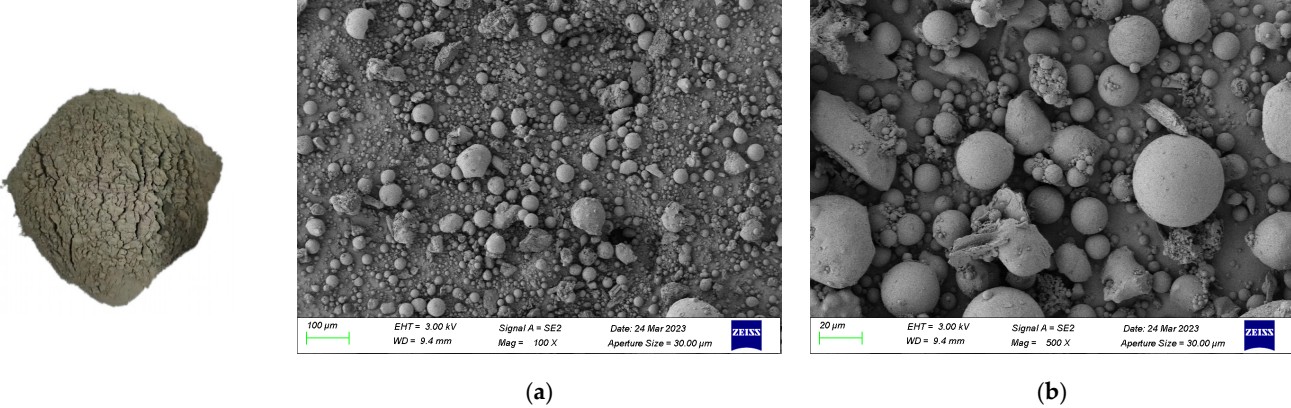

**Figure 11.** Fly ash and SEM images. (**a**) 100×; (**b**) 500×.

For the convenience of description and calculation, the total mass of coal slime was fixed at 100% and an orthogonal table with four factors at three levels was set to design test schemes, as listed in Table 7. In Table 7 and subsequent papers, A, B, C and D separately represent the coal slime, high-water-content material, cement, and fly ash.

**Table 7.** Factor (designed in proportion to mass).

| Levels | A. Mass Concentration of Coal Slime (A) | B. Content of High-Water-Content Material (B/A) | C. Cement Content (C/A) | D. Content of Fly Ash (D/A) |
|---|---|---|---|---|
| 1 | 0.4000 | 0.0300 | 0.04 | 0.04 |
| 2 | 0.3375 | 0.0375 | 0.08 | 0.08 |
| 3 | 0.2875 | 0.0450 | 0.12 | 0.12 |

### 3.2. Ultrasonic Experimental Design

The material prepared at the room temperature of 25 °C was poured into a standard triplet mold measuring 70.7 mm × 70.7 mm × 70.7 mm. After initial setting and demolding, the specimens were placed in a standard YH-60B curing box to be cured for 28 d. After 28 d, the specimens were removed to measure their ultrasonic p-wave velocities, followed by measurement of the uniaxial compressive strength and uniaxial shear strength. The ultrasonic p-wave velocities were measured using a HICHANCE HC-U81 concrete ultrasonic detector, and the strength was tested on an MTS electro-hydraulic servo-motor machine.

A substantial body of other relevant studies [28–30] has demonstrated that, in the testing of cohesive materials, pressure-waves exhibit faster propagation speeds and possess relatively greater penetration depth compared to SH waves and SV waves. Therefore, we selected pressure-waves as the focus of our current research.

The ultrasonic p-wave velocity was measured using the following method: a standard specimen was put between two probes coated with the coupling agent. One of the probes was used as the pulsed acoustic emission source and the other as the pulsed acoustic receiver. Results of the received pulses were displayed on the screen of a microcomputer equipped with an ultrasonic detector. After finishing all tests, the computer was used to process the measurement data. If the long axis along the height of the specimen is L and the arrival time is t, then the ultrasonic p-wave velocity V in the specimen is given by L/t, as shown in Figure 12. Each specimen was tested three times and the arithmetic mean was adopted as the final test value. The ultrasonic tester parameter settings are shown in Table 8. The test results are summarized in Table 9.

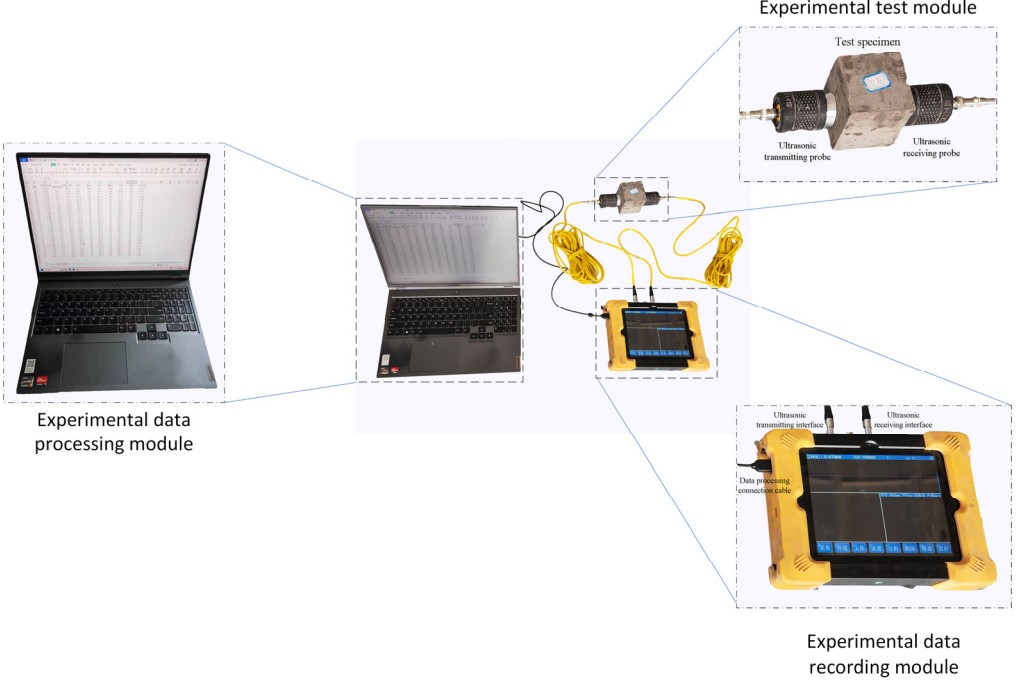

**Figure 12.** Ultrasonic p-wave velocity measurement of specimens.

**Table 8.** Ultrasonic tester parameter settings.

| | | | |
|---|---|---|---|
| Measurement point spacing (mm) | 70.7 | Sampling period (us) | 0.50 |
| Distance measurement increment (mm) | 0 | Zero sound correction (us) | 12.50 |
| Emission voltage (V) | 500 | Aggregate | none |
| Test surface | Side | Ultrasonic frequency (Hz) | 50 K |

**Table 9.** Summary of orthogonal test results.

| Test Groups | Influencing Factors | | | | Test Results | | |
|---|---|---|---|---|---|---|---|
| | **A** | **B** | **C** | **D** | **Wave Velocity/km·s⁻¹** | **Compressive Strength/MPa** | **Shear Strength/MPa** |
| 1 | 0.400 | 0.0300 | 0.04 | 0.04 | 5.992 | 0.107 | 0.056 |
| 2 | 0.400 | 0.0375 | 0.08 | 0.08 | 5.783 | 0.161 | 0.095 |
| 3 | 0.400 | 0.0450 | 0.12 | 0.12 | 8.319 | 0.234 | 0.369 |
| 4 | 0.3375 | 0.0300 | 0.08 | 0.12 | 5.205 | 0.247 | 0.109 |
| 5 | 0.3375 | 0.0375 | 0.12 | 0.04 | 3.916 | 0.086 | 0.043 |
| 6 | 0.3375 | 0.0450 | 0.04 | 0.08 | 4.706 | 0.076 | 0.052 |
| 7 | 0.2875 | 0.0300 | 0.12 | 0.08 | 5.741 | 0.115 | 0.139 |
| 8 | 0.2875 | 0.0375 | 0.04 | 0.12 | 3.936 | 0.102 | 0.036 |
| 9 | 0.2875 | 0.0450 | 0.08 | 0.04 | 4.076 | 0.105 | 0.033 |

*3.3. Experimental Data and Analysis*

The strength test is shown in Figure 13. As shown in Table 9, the p-wave velocity, compressive strength, and shear strength of the coal slime based backfill materials are 3.916~8.319 km/s, 0.076~0.247 MPa, and 0.033~0.139 MPa, respectively, each of which is distributed across a broad range. Range analysis was performed on the test results. Data in Table 10 indicate that various factors are listed in descending order as A, D, B, and C according to their influences on the p-wave velocity.

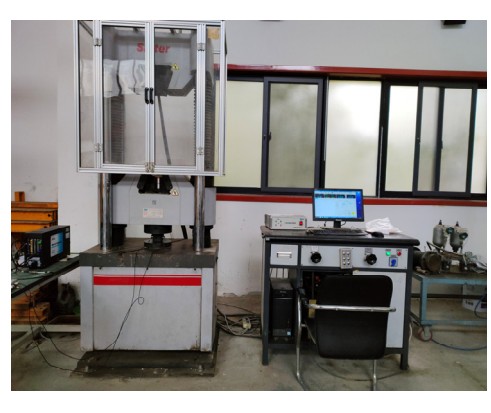
(**a**)

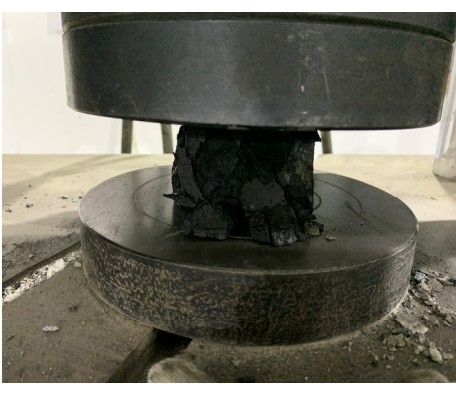
(**b**)

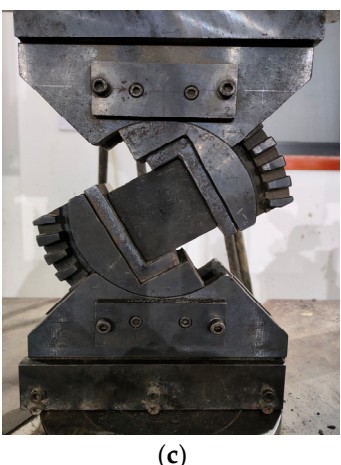
(**c**)

**Figure 13.** Strength test of specimens. (**a**) MTS electro-hydraulic servo-motor testing machine; (**b**) Compression tests; (**c**) Shear tests.

**Table 10.** Ranged analysis of test data.

| Factor | Ultrasonic P-Wave Velocity/km·s$^{-1}$ | | | |
|---|---|---|---|---|
| | **A** | **B** | **C** | **D** |
| k1 | 6.698 | 5.646 | 4.878 | 4.661 |
| k2 | 4.609 | 4.545 | 5.021 | 5.410 |
| k3 | 4.584 | 5.700 | 5.992 | 5.820 |
| R | 2.114 | 1.155 | 1.114 | 1.159 |

## 4. Correlation between Strength and P-Wave Velocity

### 4.1. Correlation between Compressive Strength and P-Wave Velocity

Variations in the ultrasonic p-wave velocity in the filling materials are correlated with the strength of the filling materials. Generally, the strength of the filling materials is related to their density and texture. The ultrasonic propagation speed in different materials is related to the material density, elastic modulus, and internal defects.

Therefore, variations in the ultrasonic p-wave velocity and strength of coal slime based backfill materials were subjected to correlation analysis. In correlation analysis, the correlation coefficient is commonly used to measure how closely variables are correlated. The most common correlation coefficient is the Pearson correlation coefficient, which measures the linear correlation between two variables. The Pearson correlation coefficient lies within the range from −1 to 1. A Pearson correlation coefficient approaching 1 or −1 indicates positive or negative correlations, while it means that there is no correlation if the coefficient approximates to 0.

As shown in Figure 14, the ultrasonic p-wave velocity varies in a manner similar to the strength of the filling materials. In the first several groups of tests, the ultrasonic p-wave velocity increases with the strength. From the third to the sixth group, the strength and ultrasonic p-wave velocity begin to decline simultaneously, and they decrease at the same time after small increments in the seventh group of tests. After correlation analysis on the ultrasonic p-wave velocity and compressive strength, a correlation heat map was drawn (Figure 15). In the figure, the color of ellipses indicates the symbol of correlation coefficients: red indicates that two factors are positively correlated, and the more oblate the ellipse is, the higher the correlation. The correlation coefficient between the ultrasonic p-wave velocity and compressive strength is 0.87, which means that they have a strong positive correlation. These results suggest that in these tests, specimens with a high (low) ultrasonic p-wave velocity generally show a high (low) compressive strength.

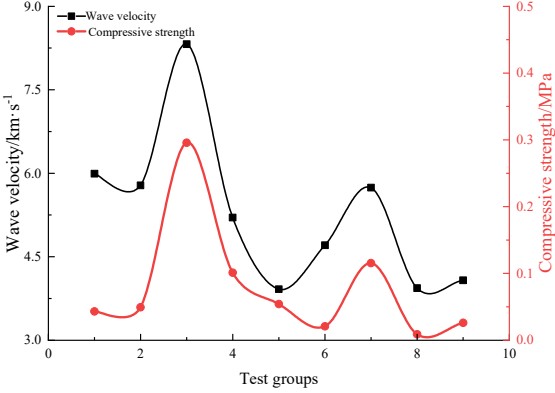

**Figure 14.** Changes in the ultrasonic p-wave velocity and compressive strength under different mix proportions.

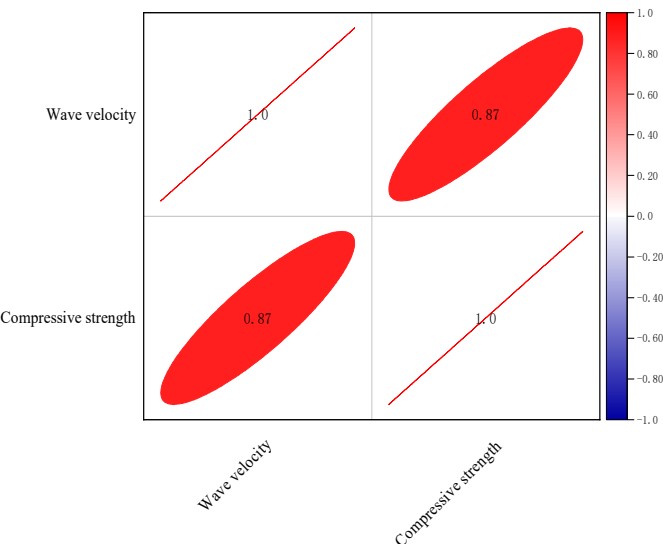

**Figure 15.** Heat map for the correlation coefficient between the ultrasonic p-wave velocity and compressive strength.

### 4.2. Correlation between Shear Strength and Ultrasonic P-Wave Velocity

Figure 16 shows that the ultrasonic p-wave velocity changes with the shear strength of the filling materials. In the first several groups of tests, the ultrasonic p-wave velocity increases with the shear strength. From the third to the sixth group, the strength and wave velocity begin to decrease at the same time; in the seventh group of tests, they decrease simultaneously after slight increments.

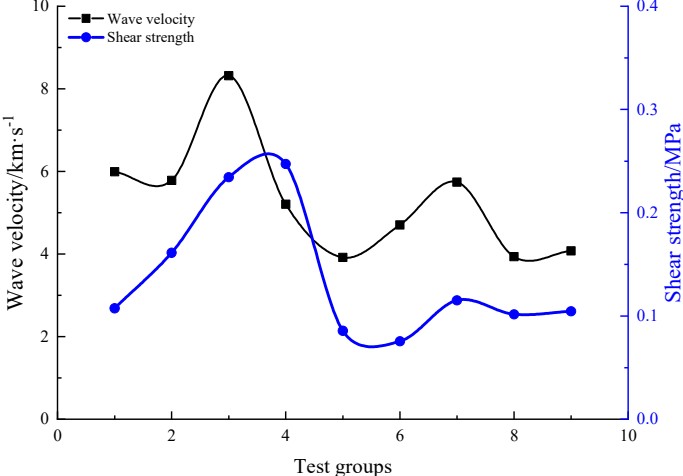

**Figure 16.** Changes in the ultrasonic p-wave velocity and shear strength under different mix proportions.

As can be seen from Figure 17, the ultrasonic p-wave velocity and the shear strength have a correlation coefficient of 0.65, indicative of a positive correlation, suggesting that, in these tests, specimens with a high (low) ultrasonic p-wave velocity generally show a high (low) shear strength.

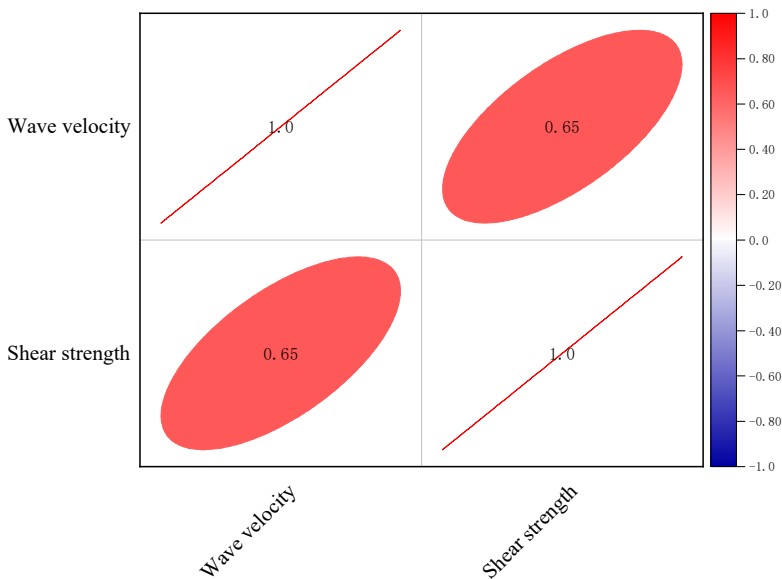

**Figure 17.** Heat map for the correlation coefficient between the ultrasonic p-wave velocity and shear strength.

## 5. Influences of Mix Proportions of Materials on the Ultrasonic P-Wave Velocity

Based on orthogonal test results, influences of multiple factors on the ultrasonic p-wave velocity are displayed in Figure 18. Various factors are found to differ in their influences on the ultrasonic p-wave velocity. According to analysis of test results, the content of fly ash was listed in the error row and significance testing was conducted at each level. The results of analysis of variance (ANOVA) are listed in Table 11.

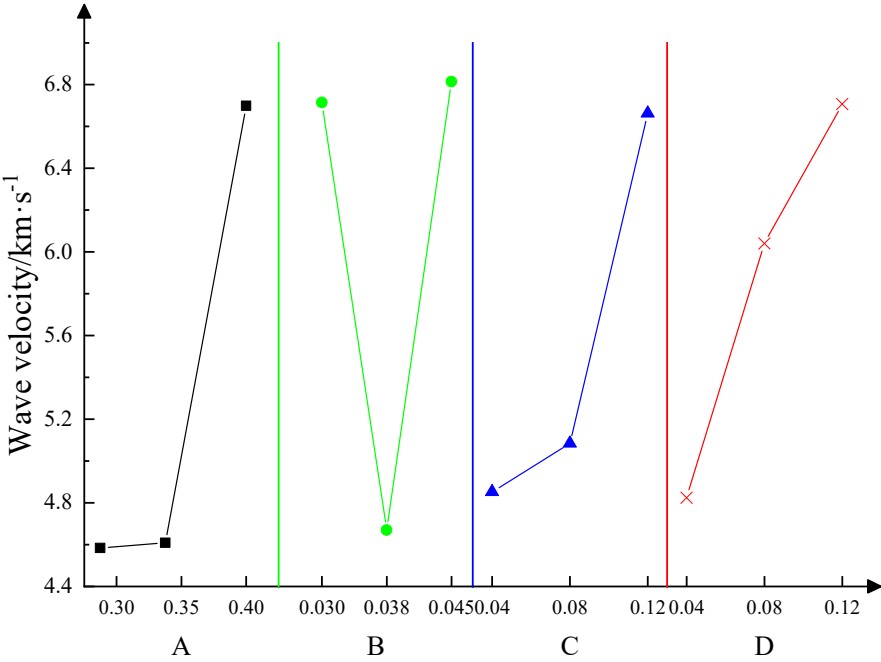

**Figure 18.** Relationship between the ultrasonic p-wave velocity and multiple factors. Different color lines in the figure represent different materials A, B, C and D.

**Table 11.** Analysis of variance of test data.

| Source of Variation | Sum of Squared Deviations | Degree of Freedom | Variance | F-Value | Fα [2] | Significant Level |
|---|---|---|---|---|---|---|
| A | 8.8312 | 2 | 4.4156 | 3.884 | | [1]* |
| B | 2.5488 | 2 | 1.2744 | | F0.01(2,6) = 10.925 | |
| C | 2.2027 | 2 | 1.1013 | | F0.05(2,6) = 5.143 | |
| Error e | 2.0701 | 2 | 1.0351 | | F0.1(2,6) = 3.463 | |
| Correction error e | 6.8216 | 6 | 1.1369 | | F0.25(2,6) = 1.762 | |
| Sum | 15.653 | | | | | |

[1] "*" in the table means that the factor has a certain impact on the wave speed. [2] The symbol Fα represents the critical values of the F-statistic for degrees of freedom of 2 in the numerator and 6 in the denominator, at significance levels of 1%, 5%, 10%, and 25%. These critical values are used in hypothesis testing in analysis of variance to help ascertain if the differences between groups are significant.

In the ANOVA, due to the small variance of parameter D, by treating D as an error term, its influence on the dependent variable is reduced. This also reduces the variability of the experimental results, thereby enhancing the accuracy and reliability of the experiment.

ANOVA shows that, within a given range, factor A significantly influences the ultrasonic p-wave velocity, while factors B and C exert slight influences. The values of variance suggest that factor B exerts a greater influence than factor C, and the mass concentration of coal slime is the main factor affecting the ultrasonic p-wave velocity of specimens.

## 6. Parametric Analysis Model of Ultrasonic P-Wave Velocity

*6.1. Variation of Characteristic Parameters under the Influences of Single Factors*

Influences of various factors on the ultrasonic p-wave velocity of specimens were revealed through the above tests with different mix proportions. The results are displayed in Figure 19. The ultrasonic p-wave velocity tends to decrease, and then increase with the increments of factors A and B, while it increases with factors C and D.

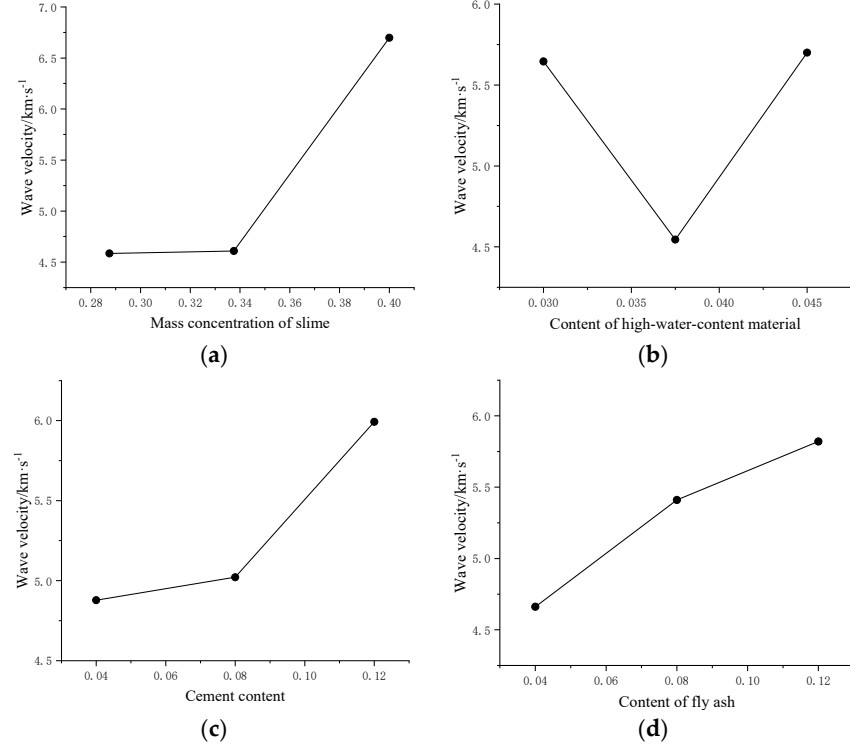

**Figure 19.** Ultrasonic p-wave velocities of materials under influences of different factors. (**a**) The effects of coal slime; (**b**) The effects of high-water-content material; (**c**) The effects of cement; (**d**) The effects of fly ash.

### 6.2. Parametric Predictive Model

Various influencing factors and parametric curves of materials in Figure 13 were fitted. In this way, the relationships of the mass concentration of coal slime, content of high-water-content material, cement content, and content of fly ash with the ultrasonic p-wave velocity were acquired (Table 12).

**Table 12.** Fitting relationship.

| Factor | Index | Fit Relation | General Formula | Degree of Fit $R^2$ |
|---|---|---|---|---|
| Mass concentration of coal slime($P_A$) | Wave velocity | $v_1 = 292.71704P_A{}^2 - 182.45481P_A + 32.8452$ | $v_1 = aP_A{}^2 + bP_A + c$ | 0.99 |
| High-water content($P_B$) | Wave velocity | $v_2 = 5.67317 - 1.12815e^{-\frac{(P_B - 0.0375)^2}{2 \times 0.00299^2}}$ | $v_2 = y_0 + Ae^{-\frac{(P_B - x_c)^2}{2w^2}}$ | 0.99 |
| Cement content($P_C$) | Wave velocity | $v_3 = 4.85317 + 0.02483e^{(P_D - 0.04)/0.02091}$ | $v_2 = y_0 + A_1e^{(P_C - x_0)/t_1}$ | 0.99 |
| Fly ash content($P_D$) | Wave velocity | $v_4 = 6.31636 - 3.0221e^{-P_D/0.06643}$ | $v_2 = y_0 + A_1e^{-P_D/t_1}$ | 0.99 |

The data show that the ultrasonic p-wave velocity has a quadratic polynomial relationship with factor A, while it follows an exponential relationship with factors B, C, and D. The fitted curves of the four factors were plotted, as illustrated in Figure 20.

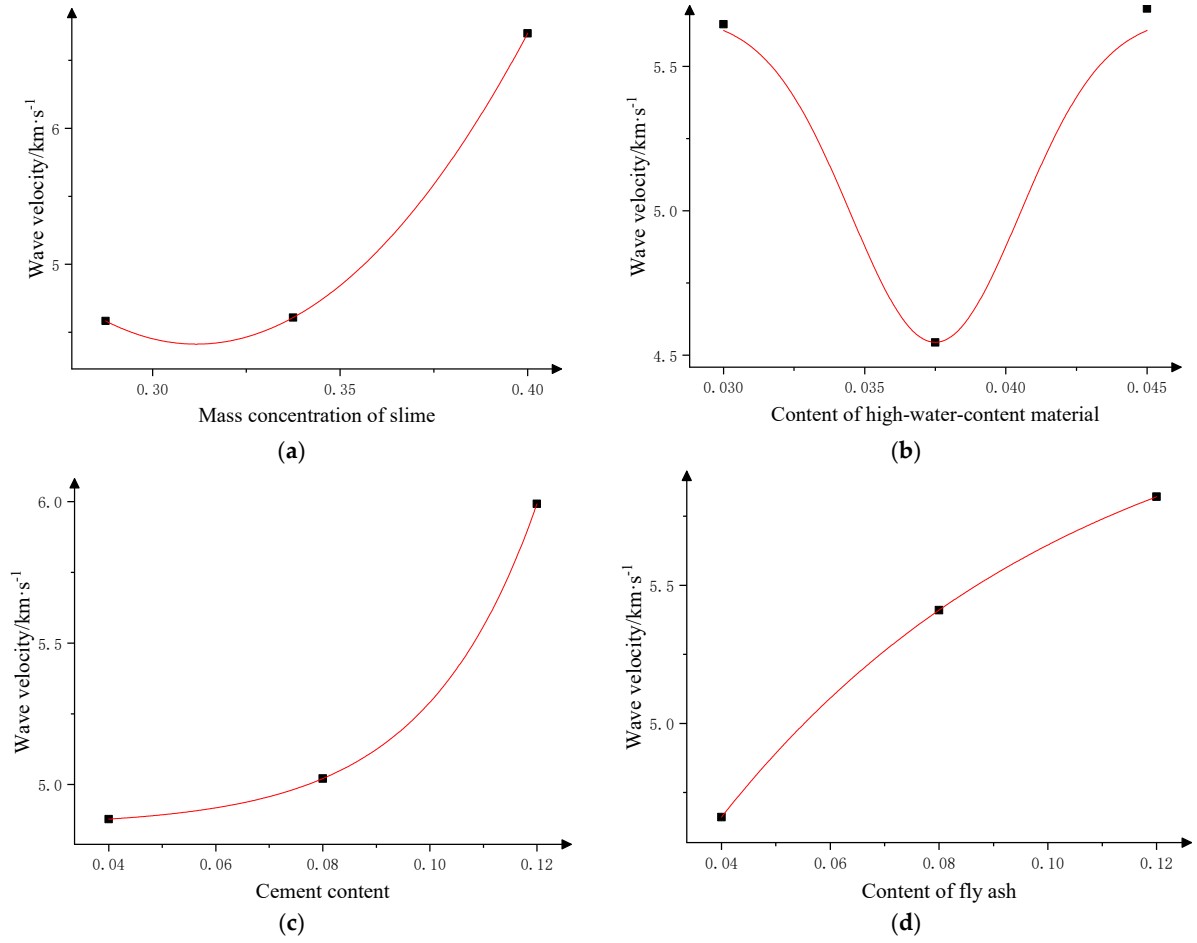

**Figure 20.** For the wave velocity fitting curves of different factors. (**a**) The effects of coal slime; (**b**) The effects of high-water-content material; (**c**) The effects of cement; (**d**) The effects of fly ash.

### 6.3. Establishment of the Predictive Model

According to relationships of the ultrasonic p-wave velocity with various factors, a synthetic forecasting model for characteristic parameters of the material was constructed combining with the principle for developing multiple linear models, as expressed by Formula (1).

$$\{v = a_1v_1 + a_2v_2 + a_3v_3 + a_4v_4 + a_5 \tag{1}$$

According to numerous test results pertaining to each of the characteristic parameters, Equation (1) is subjected to multiple linear regression, thus acquiring the relationships of the ultrasonic p-wave velocity with various factors, as expressed by Formula (2).

$$\{v = 0.97093v_1 + 1.007324v_2 + 0.95037v_3 + 0.99147v_4 - 15.43465 \tag{2}$$

The independent variable interval of the fitted wave velocity prediction equation is $0.2875 < v_1 < 0.4000$; $0.0300 < v_2 < 0.0450$; $0.04 < v_3 < 0.12$; $0.04 < v_4 < 0.12$. In this interval, the predicted equation fits the actual measured wave velocity to a high degree.

### 6.4. Verification of the Synthetic Forecasting Model

Formula (2) was verified: the filling material slurry and specimens were prepared by selecting two groups of mix proportions (Table 13), the ultrasonic p-wave velocity of which was measured. The predicted values obtained by using the predictive model were compared with the test values (comparisons of results and errors are listed in Table 14).

**Table 13.** Design of verification schemes.

| The Specimen Number | A. Mass Concentration of Coal Slime | B. Content of High-Water-Content Material | C. Cement Content | D. Content of Fly Ash |
|---|---|---|---|---|
| C1 | 0.35 | 0.037 | 0.06 | 0.10 |
| C2 | 0.31 | 0.039 | 0.07 | 0.11 |
| C3 | 0.32 | 0.041 | 0.10 | 0.05 |

**Table 14.** Analysis of verification results.

| The Specimen Number | Materials | Wave Velocity/km·s$^{-1}$ |
|---|---|---|
| C1 | Predicted value | 4.133 |
| | Experimental value | 4.324 |
| | Error/% | 4.413% |
| C2 | Predicted value | 3.965 |
| | Experimental value | 4.231 |
| | Error/% | 6.270% |
| C3 | Predicted value | 3.891 |
| | Experimental value | 4.051 |
| | Error/% | 3.950% |

Analysis unveils that the parameters calculated using the synthetic forecasting model conform well to the test values. The relative errors of various indices are in a range of 4% to 6%, with an average value of 4.878%.

## 7. Conclusions

(i) Orthogonal tests of the ultrasonic p-wave velocity were conducted on four factors including the mass concentration of coal slime, content of high-water-content material, cement content, and content of fly ash that were set at three levels using the orthogonal test method. Afterwards, analysis shows that the ultrasonic p-wave velocity of the coal slime based backfill materials is in the range of 3.916~8.319 km/s.

(ii) The wave velocity was found to have a positive correlation with compressive strength, with a correlation coefficient of 0.87. It also exhibited a positive correlation with

shear strength, with a correlation coefficient of 0.65. Various factors are listed in descending order as A, D, B, and C according to their influences on the ultrasonic p-wave velocity, suggesting that the mass concentration of coal slime and content of fly ash are main factors influencing the ultrasonic p-wave velocity.

(iii) The mathematical relationships of factors A, B, C, and D with the ultrasonic p-wave velocity were fitted. The wave velocity has a quadratic polynomial relationship with factor A while being governed by exponential relationships with factors B, C, and D. The synthetic forecasting model was obtained through the multiple regression equation, and the ultrasonic p-wave velocity calculated thereby has an average error of 4.878% with the test data.

**Author Contributions:** Supervision, B.A. and J.R.; Conceptualization, B.A.; Resources, B.A.; Writing—original draft, C.C.; Writing—review and editing, B.A. and J.W.; Formal analysis, C.C. and D.W.; Investigation, C.C., D.W. and Q.Y.; Funding acquisition, B.A. and J.R.; Data curation, D.W. and C.C.; Validation, J.W. All authors have read and agreed to the published version of the manuscript.

**Funding:** This research was funded by the National Natural Science Foundation of China (NO. 52274117 and NO. 52174128).

**Institutional Review Board Statement:** Not applicable.

**Informed Consent Statement:** Not applicable.

**Data Availability Statement:** Not applicable.

**Conflicts of Interest:** The authors declare no conflict of interest.

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
