# Peer review of "Experimental Research into the Ultrasonic P-Wave Velocity of Coal Slime Based Backfill Material"

_applsci, doi:10.3390/app131911043_

Round 1
Reviewer 1 Report
After careful reading of this interesting manuscript, I have the following doubts, namely:
1. It is well known that in the case of 3D waves, we have three elastic wave mods, pressure-waves (p-waves), shear-horizontal waves (SH-waves), and shear-vertical waves (SV-waves). In the manuscript, there is no information about the SH and SV wave modes and whether they interact with the p-wave mod or whether they are present at all. How the authors extract the p-wave mode from the registered dynamic response of the specimens.
2. I am not very familiar with this kind of material but in the case of most elastic wave modes the dispersion phenomenon is present. This phenomenon makes the wave speed measurement very difficult. In the manuscript, there is no information or graphs, which represent the input signal and the received signal of acoustic emission. There is no information about the dispersion properties of the studied p-wave (The dependency between the frequency and the wave speed).
3. There is no information about the way the wave speed is estimated. As mentioned above, if the analyzed p-wave poses a dispersive character, the accurate wave speed measurement could be very difficult. For example, we need to determine the group velocity and based on it to try evaluate the wave speed. In the manuscript, there is no information on how exactly the wave speed measurement (phase or group) is carried out.
Author Response
Thank you very much for your advice. I have made revisions one by one based on your suggestions.
1.It is well known that in the case of 3D waves, we have three elastic wave mods, pressure-waves (p-waves), shear-horizontal waves (SH-waves), and shear-vertical waves (SV-waves). In the manuscript, there is no information about the SH and SV wave modes and whether they interact with the p-wave mod or whether they are present at all. How the authors extract the p-wave mode from the registered dynamic response of the specimens.
A substantial body of other relevant studies[1-3] has demonstrated that, in the testing of cohesive materials, pressure-waves exhibit faster propagation speeds and possess relatively greater penetration depth compared to SH waves and SV waves. Therefore, we selected pressure-waves as the focus of our current research. Specifically, we concentrated solely on the arrival time of the first wave and the time taken for the first wave to traverse the specimen and be received on the opposite side.
In our experiments, we employed high-precision and highly stable ultrasonic testing equipment along with sensors. The ultrasonic equipment utilized for this experiment featured two sensors - a flat transducer (one for ultrasonic wave generation and the other for signal reception). The performance of these devices ensured our ability to obtain accurate input and received signals. This allowed us to accurately record the occurrence time of the first longitudinal wave generated by one side's flat transducer and the time it was received by the flat transducer on the opposite side with high precision.
Moreover, our test specimens were in the shape of cubes (with dimensions of 70.7×70.7×70.7mm), and the two flat transducers were symmetrically positioned (as illustrated in the diagram below). This configuration ensured that the path between the two symmetric faces was the shortest, enabling the pressure-wave to reach the transducer first and resulting in clear waveform signals being captured.
- Li, H.G.; Wang, J.; Liu, K. Study on the relationship between longitudinal wave velocity and mechanical properties of overlying rock in Shendong Mining area. J. Mining Research and Development, 2023, 43(08):113-119. DOI:10.13827/j.cnki.kyyk.2023.08.006.
- Zhao, M.J.; Wu, D.L.; Ultrasonic classification and strength prediction of engineering rock mass. J. Journal of Rock Mechanics and Engineering. 2000(1):89-92.
- You, M.Q.; Su, C.D.; Li, X.S. Study on the relationship between mechanical properties of damaged rock samples and longitudinal wave velocity. Journal of Rock Mechanics and Engineering. 2008(3):458-467
- I am not very familiar with this kind of material but in the case of most elastic wave modes the dispersion phenomenon is present. This phenomenon makes the wave speed measurement very difficult. In the manuscript, there is no information or graphs, which represent the input signal and the received signal of acoustic emission. There is no information about the dispersion properties of the studied p-wave (The dependency between the frequency and the wave speed).
Our specimens are in the shape of cubes with dimensions of 70.7×70.7×70.7mm. The two flat transducers are symmetrically positioned, resulting in the shortest path between two symmetric faces. As a result, the first longitudinal wave reaches the transducer, and the captured waveform signal is clear. Both shear waves and surface waves arrive after this signal.
Furthermore, the raw materials of our specimens are composed of finely ground particles that are mixed and bonded, and the production process includes the use of a vibrating table to eliminate any air bubbles within the specimen. This ensures a compact and uniform overall structure, preventing significant scattering and signal attenuation caused by internal bubbles when the longitudinal wave encounters them.
We have included additional information about the longitudinal wave signals in the manuscript. Please refer to line 286 of the paper.
3.There is no information about the way the wave speed is estimated. As mentioned above, if the analyzed p-wave poses a dispersive character, the accurate wave speed measurement could be very difficult. For example, we need to determine the group velocity and based on it to try evaluate the wave speed. In the manuscript, there is no information on how exactly the wave speed measurement (phase or group) is carried out.
Our specimens are in the shape of cubes with dimensions of 70.7×70.7×70.7mm. The two flat transducers are symmetrically positioned, resulting in the shortest path between two symmetric faces. As a result, the first longitudinal wave reaches the transducer, and the captured waveform signal is the clearest.
Furthermore, the raw materials of our specimens are composed of finely ground particles that are mixed and bonded. The production process includes the use of a vibrating table to eliminate any air bubbles within the specimen. This ensures a compact and uniform overall structure, preventing significant scattering and signal attenuation caused by internal bubbles when the longitudinal wave encounters them.
During the process of measuring wave speeds, the scattering phenomenon is relatively weak. Due to this, the testing equipment only recorded data for the first wave, which is the primary focus of our study.

Reviewer 2 Report
The most important change is the word "slime" which does not convey that the author proposes recycling of waste material produced during mine exploitation in order to use it as backfill for mine rehabilitation. Other issues concern the research presentation and the results and can be found as comments in the attached review.

The english is generally clear.
Author Response
Thank you very much for your advice. I have made revisions one by one based on your suggestions.
1.coal tailings based backfill material. Change the word slime throughout the entire document
We have checked and modified the corresponding words in the article.
2.Provide a concise description of how these waste materials are produced and what substances they may contain.
The relevant content has been added to the 74 line of the paper.
We have supplemented the paper with relevant descriptions and analyses of the physical and chemical properties of the materials used, including coal slime, fly ash, cement, and high-water-materials."
3.Provide a chemical analysis of the material so as to make the research repeatable. Similarly for the rest of the materials, eg. what is the "high water content"material? be specific, provide precise information.
We have supplemented the paper with relevant descriptions and analyses of the materials used, including coal slurry, fly ash, cement, and high-water materials. For details, please refer to table 1-6 of the paper.
4.Also provide what frequency / wavelenth were the pulses.
Based on the suggestions, we have added the corresponding parameter data. For details, please refer to Table 8 in the paper.
5.Unclear explanation of ANOVA results, the table contains F-value only for A, column Fa is not explained, parameter D is not included. No insights and reasoning for what the statistical analysis points to are provided.
We have provided supplementary explanations for the aforementioned parameters. For detailed information, please refer to line 346 of the paper.
Fα is a statistical measure used to determine whether there are significant differences in variance among different groups in a model. The symbol α represents the critical values of the F-statistic for degrees of freedom of 2 in the numerator and 6 in the denominator, at significance levels of 1%, 5%, 10%, and 25%. These critical values are used in hypothesis testing in analysis of variance to help ascertain if the differences between groups are significant. If the computed F-statistic exceeds the corresponding critical value, we can reject the null hypothesis and conclude that there are significant differences among the groups.
In the analysis of variance, due to the small variance of parameter D, by treating D as an error term, its influence on the dependent variable is reduced. This also reduces the variability of the experimental results, thereby enhancing the accuracy and reliability of the experiment. However, this effect is not directly reflected in the analysis of variance.
6.The limits of the proposed equation must be added, i.e. the domain for selecting values for the 4 parameters but be limited within the values of the research.
We have supplemented the mentioned parameter ranges. Please refer to line 385 of the paper for specific details.
The independent variable ranges for the fitted wave speed prediction equation are 0.2875<v1<0.4000; 0.0300<v2<0.0450; 0.04<v3<0.12; 0.04<v4<0.12. Within this range, the prediction equation shows a high degree of fit with the actually measured wave speeds."

Round 2
Reviewer 1 Report
In my opinion, the answers to the point 2 and 3 of my review are still not satisfactory. Still, there is no information about the depression of elastic waves and there is no precise information about how the p-wave velocity is estimated. The analysis of the character of the wave propagation possesses fundamental meaning. In my opinion, it is not clear what is shown in Figure 13. Is this the dynamic response of the structure received by the sensors ?! Generally, there is no information about the elastic wave propagation phenomenon. It is interesting how much the velocity speed of the modes SH and SV is less in comparison with the pressure wave.
Please, show me at least only one really registered signal by the sensors.
Additionally, the answer to remark 3 seems not to be off-topic. There is still no information on how the -p-wave speed is estimated (phase of group velocity). If the character of the wave propagation is unknown ( dispersive or not), a reliable estimation of the wave propagation is impossible.
There is a fundamental difference between the dispersion and scattering phenomena of the waves.
Author Response
Thank you very much for the valuable suggestions. In response to the issues you raised, our team has conducted a thorough discussion,and the reply is attached.
